# Association between long COVID and vaccination: A 12-month follow-up study in a low- to middle-income country

Samar Fatima[1]☯, Madiha Ismail📷[2]☯*, Taymmia Ejaz📷[1]‡, Zarnain Shah[1]‡, Summaya Fatima[1]‡, Mohammad Shahzaib[1]‡, Hassan Masood Jafri[2]‡

1 Section of Internal Medicine, Department of Medicine, Aga Khan University Hospital, Karachi, Pakistan,
2 Department of Emergency Medicine, Aga Khan University Hospital, Karachi, Pakistan

☯ These authors contributed equally to this work.
‡ TE, ZS, SF, MS and HMJ also contributed equally to this work.
* madiha.ismail@aku.edu

## Abstract

### Objective

There is a lack of estimates regarding the at-risk population associated with long COVID in Pakistan due to the absence of prospective longitudinal studies. This study aimed to determine the prevalence of long COVID and its association with disease severity and vaccination status of the patient.

### Design and data sources

This prospective cohort study was conducted at the Aga Khan University Hospital and recruited patients aged > 18 years who were admitted between February 1 and June 7, 2021. During this time, 901 individuals were admitted, after excluding patients with missing data, a total of 481 confirmed cases were enrolled.

### Results

The mean age of the study population was 56.9±14.3 years. Among patients with known vaccination status (n = 474), 19%(n = 90) and 19.2%(n = 91) were fully and partially vaccinated, respectively. Severe/critical disease was present in 64%(n = 312). The mortality rate following discharge was 4.58%(n = 22). Around 18.9%(n = 91) of the population required readmission to the hospital, with respiratory failure (31.8%, n = 29) as the leading cause. Long COVID symptoms were present in 29.9%(n = 144), and these symptoms were more prevalent in the severe/critical (35.5%, n = 111) and unvaccinated (37.9%, n = 105) cohort. The most prominent symptoms were fatigue (26.2%, n = 126) and shortness of breath (24.1%, n = 116), followed by cough (15.2%, n = 73). Vaccinated as compared to unvaccinated patients had lower readmissions (13.8% vs. 21.51%) and post-COVID pulmonary complications (15.4% vs. 24.2%). On multivariable analysis, after adjusting for age, gender, co-morbidity, and disease severity, lack of vaccination was found to be an independent predictor of long COVID with an Odds ratio of 2.42(95% CI 1.52–3.84). Fully and partially vaccinated patients had 62% and 56% reduced risk of developing long COVID respectively.

**Data Availability Statement:** All relevant data are within the paper and its Supporting Information files.

**Funding:** The authors received no funding for their work.

**Competing interests:** All authors have declared that no competing interest exists.

## Conclusions

This study reports that the patients continued to have debilitating symptoms related to long COVID, one year after discharge, and most of its effects were observed in patients with severe/critical disease and unvaccinated patients.

## Introduction

Emerging evidence indicates that COVID-19 has long-term consequences on the immunological, respiratory, neuropsychiatric, cardiac, haematological, and functional abilities of patients [1–3]. Although acute damage to multiple organs has already been established in this disease, the long-term effects of this disease need to be considered [4]. Around 5% of individuals with COVID-19 experience a severe form of the illness that necessitates hospitalization in an intensive care unit (ICU), and approximately two-thirds of these individuals develop acute respiratory distress syndrome (ARDS), with only 25% surviving the illness [5]. Severe and critical forms of the disease that develop ARDS during hospital admission can lead to a disorder characterized by persistent fatigue, weakness, and limited exercise tolerance [6]. These patients often have sequelae from their illness and hospital stay, which impair their overall health status and create significant health needs after hospitalization. The occurrence of debilitating, ongoing symptoms of COVID-19 is common. Even those with milder infections have reported persistent problems. They belong to a vulnerable population, and therefore the burden of care for this population is suspected to be substantial.

While research efforts have been expedited to address treatment and vaccination for preventing transmission and mortality, there has been a notable lack of research in areas such as diagnostic criteria, establishing a consistent definition, understanding the pathophysiology, and developing effective strategies for managing and treating long COVID. Globally, it has been estimated that at least 65 to 144 million individuals may have developed long COVID by the end of 2021 [7]. Based on 3.92 billion SARS-COV-2 infections by 2021, with 3.7% of these reporting long COVID symptoms, the Institute for Health Metrics and Evaluation (IHME) conducted disease modelling for the estimation of long COVID cases. Bayesian meta-regression used data from 54 studies and 2 record databases on 1.2 million patients from 22 countries and estimated that 144.7 million (95% CI 54.8–312.9) people suffered from any of the three symptom clusters of long COVID [8], with 15.1% of the individuals having persistent symptoms at 12 months, that comes to an estimated burden of 21 million individuals suffering from long COVID. The economic, social, and psychological impact of long COVID has also been huge. In the United States alone, the economic ramifications of long COVID are estimated to amount to approximately $170 billion in lost wages. In a large-scale survey, 18% of patients who had full-time employment before COVID, could not return to work due to long COVID symptoms [9]. It has been observed that low antibody titers to SARS-CoV-2 have been associated with a greater likelihood of experiencing long COVID, regardless of hospitalization status [10]. Perlis et al. conducted a study and observed that full vaccination resulted in a lower risk of developing long COVID with an OR of 0.72 [11].

While studies on the impact of vaccination using real-time surveillance, have reported a reduction in hospitalization and mortality [12], the long-term impact of vaccination and the number of doses or booster doses in reducing long COVID symptoms has not been widely studied, particularly in lower-middle-income countries (LMICs). Due to a lack of standardized measures, the actual extent of the disease burden remains uncertain. As per Fan et al., the

global burden of COVID during the years 2020–2021 was found to be 31,930,000 DALYs (Disability-adjusted Life Years) [13, 14]. Furthermore, studies are required on long COVID prevalence, clinical presentations, waning, or improvement in symptoms over time, and the factors that predict these outcomes. This research is crucial for understanding the overall burden, economic consequences, healthcare planning, and facilitating the return to employment for affected individuals.

To the best of our knowledge, the characteristics and long-term outcomes of COVID-19 survivors discharged from hospitals to home settings in LMICs, especially those with severe/critical disease, along with the effect of vaccines on their subsequent health, are yet to be ascertained. Therefore, this study was conducted to prospectively investigate the long-term sequelae of COVID-19 infection on symptoms, mental health, and functional recovery according to the disease severity and vaccination status after one year following discharge from the hospital. This information will offer insights into the current physical and mental health status of the patient population, enabling us to develop tailored rehabilitation programs for those who have survived the pandemic while being affected by severe and critical forms of the disease.

## Methods

### Study design/data source

This prospective cohort longitudinal follow-up study was conducted at Aga Khan University Hospital (AKUH). As the largest tertiary care centre in Pakistan, AKUH serves patients from across the nation and offers comprehensive treatment for a wide spectrum of diseases and conditions regardless of their severity. AKUH is a 740-bed hospital with fully equipped emergency rooms, well-appointed critical care units, and specialized wards, all aimed at maintaining a high standard of quality care for all admitted patients. In response to the COVID-19 pandemic, AKUH allocated a separate building for the treatment of COVID-19 patients and has, to date, admitted and provided care for more than 5,200 individuals affected by the virus.

### Eligibility criteria and data collection

Adult patients admitted between February 1 and June 7, 2021, who had at least one positive SARS-CoV-2 RT-PCR result in a nasopharyngeal/oropharyngeal swab/tracheal sample were included in the study. Patients with symptoms suggestive of COVID-19 infection but negative RT-PCR results were excluded from the study.

During this time, 901 individuals were admitted to AKUH intensive care/high dependency units and wards with confirmed COVID-19 infection. The in-hospital mortality was 11.5%. After excluding the patients with missing data, a total of 481 confirmed cases admitted to AKUH were recruited, medical records reviewed, and data recorded in predesigned proforma. Around twenty-two patients (4.58%) died after discharge from the hospital, however, it was difficult for us to assess the cause of death through telephonic interviews. Patients who could not be contacted by phone and those who refused to participate in the study were excluded, see **Fig 1**.

Patients were contacted via phone 1 year after the discharge. After telephonic consent, the research assistant utilized a standard, pre-designed questionnaire to collect the outcome data. The questionnaire inquired about details of daily activities, current health, and symptoms such as dyspnoea with exertion as defined by the New York Heart Association (NYHA) class. General symptoms referring to the survivors' overall discomfort, including physical decline or fatigue, cough, shortness of breath, and other respiratory symptoms, including chest tightness, wheezing, and chest pain were also noted. Depression and anxiety were assessed using a validated Hospital Anxiety and Depression Scale (HADS) questionnaire and were noted on the preformed questionnaire. The HADS consists of 14 questions and is divided into two

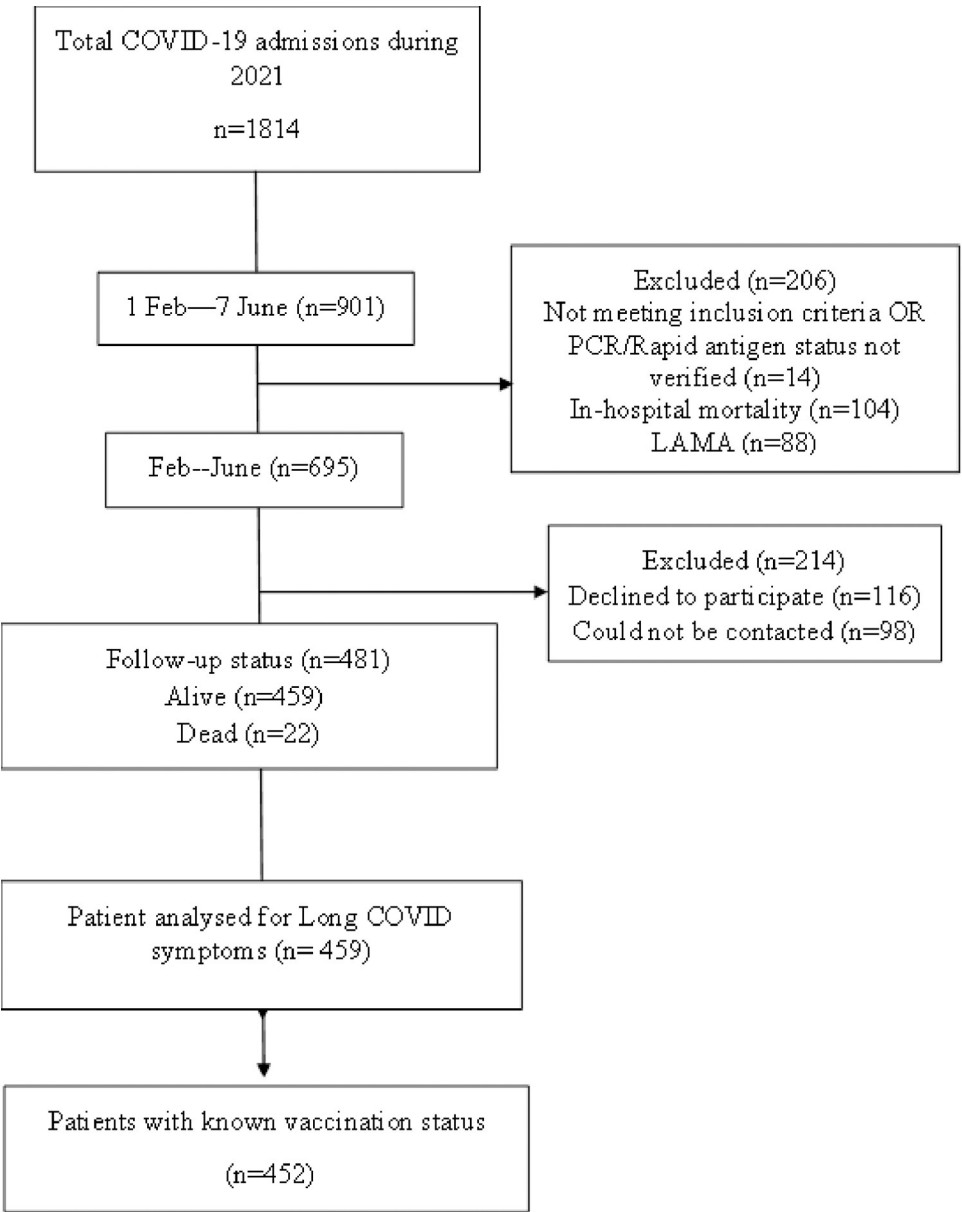

**Fig 1. STROBE patient selection flow chart.** STROBE, Strengthening the Reporting of Observational Studies in Epidemiology.

subscales: depression and anxiety. Each response has a four-point rating, with the highest score for anxiety and depression being 21. Scores of 11 or above on either subscale indicate a major "case" of psychological morbidity, whereas 8–10 indicate "borderline" and 0–7 indicate "normal" mental health [15].

Disease severity was defined as non-severe, severe, and critical according to the National Institute of Health (NIH) Pakistan guidelines. The non-severe disease was defined as patients with no symptoms or only mild symptoms with room air SpO2 of greater than 94%. The patient was classified as having severe disease when the SpO2 was below 94% on room air or when the chest X-ray findings showed lung infiltrates >50% and critical disease when they had respiratory failure requiring invasive or non-invasive ventilation (NIV).

Patients were classified according to their vaccination status into three categories: fully vaccinated, partially vaccinated, and unvaccinated. An individual was considered as "fully vaccinated" when they became symptomatic after 14 days of receiving the second dose of the vaccine, however, the patient was considered as being "partially vaccinated" when they became symptomatic ≥ 2 weeks after receiving the first dose, did not receive the second dose, or became symptomatic ≤ 2 weeks after receiving the second dose. Those who did not receive any vaccine dose were labelled as "unvaccinated" [12]. According to World Health Organization (WHO) guidelines, long COVID or post-COVID condition (PCC) was defined as the persistence of existing symptoms or the emergence of new symptoms at least three months after the first SARS-CoV-2 infection and lasting two months with no other explanation [16]. These terms are also interchangeable with post-acute COVID-19 syndrome or persistent post-COVID-19 syndrome.

## Patient and public involvement statement

This prospective study was conducted by reviewing medical charts and electronic data. Patients were interviewed by phone after obtaining informed verbal consent. Patient confidentiality and anonymity were maintained. No identifiers that could be used to track participants were used, and the research questionnaire was identified by a serial number. The study was approved by the ethical review committee (ERC) of AKUH, Karachi, Pakistan (ERC reference number: 2021-6336-19452).

## Statistical analysis

Data were entered and analyzed in Statistical Package for Social Sciences (SPSS) version 25.

First, the descriptive statistics were calculated. The normality of the data was checked using the Shapiro–Wilk test. The mean± standard deviation (SD) was calculated for quantitative continuous symmetric data, whereas median and intra quartile range (IQR) were calculated for quantitative continuous skewed data such as readmission number of visits and length of stay. Frequencies and percentages were calculated for the categorical data. For the comparison of categorical variables, the chi-square test was applied for parametric data, while Fisher's exact was applied for non-parametric data. Independent samples student t-test and Mann-Whitney U Test were applied for parametric and non-parametric continuous variables, respectively. Multi-variable logistic regression analysis was conducted to evaluate independent predictors of long COVID. A p-value of ≤0.05 was considered significant.

## Results

### Baseline characteristics of the study subjects

In total, 481 admissions were included in this study. The mean age of the study population was 56.9±14.3 (range 22–94) years. The majority were males 61.7% (n = 297), and the most common comorbid condition was diabetes mellitus (55%, n = 213) followed by hypertension (26%, n = 101). Among the study population, we were able to determine the vaccination status of 474 patients, with 19% (n = 90) and 19.2% (n = 91) fully and partially vaccinated, respectively; however, many of the admitted population were unvaccinated (61.8%, n = 293). Sixty-four percent (n = 312) of the individuals had severe/critical disease, and 25.6%(n = 120) required non-invasive (NIV)/invasive mechanical ventilation (IMV). The median length of hospital stay at the first admission was 4 (2–7) days, with a prolonged length of stay in patients with critical disease (median 8, 6–11 days). Following discharge, 18.9% (n = 91) of the population required readmission to the emergency department, with 13.9% (n = 67) requiring inpatient

hospitalization and 65.6% (n = 44) being admitted to a Special Care Unit (SCU)/Intensive Care Unit (ICU). Around 28.3% (n = 136) of patients had their chest X-rays done in the OPD (at any point from their hospital discharge up to the one-year interview). Fibrosis and consolidation were observed in 8.7% (n = 42) and 7.7% (n = 37) of the patients respectively, the rest of the chest X-rays were reported as normal. The mortality rate following discharge from the hospital after the initial/index admission was around 4.58% (n = 22). The patients who expired following their discharge had a median survival duration of 31.5 days (IQR 7.5–213), see **Table 1**.

## Reasons for readmission of COVID-19 survivors on follow-up

Among the 91 patients who required readmission, the leading complaints/diagnoses were related to respiratory involvement (31.8%, n = 29) and infections (17.6%, n = 16), followed by cardiovascular-related impairment in 12%(n = 11) of cases. Respiratory failure (27.4%, n = 25) and infection (17.1%, n = 17) were the most common diagnoses on readmission. Many of the readmitted patients had more than one diagnosis and more than one visit to the ER/hospitalization. Nonetheless, we did not find long COVID as a contributing factor for the readmissions, see **Table 2.**

## COVID-19 survivors' health-related characteristics stratified according to vaccination status

Severe disease (including critical) was found to be higher in unvaccinated patients as compared to the vaccinated cohort (68.2% vs 58.5%), with the critical disease in 27.3% (n = 80) and 20.4% (n = 37) of the unvaccinated and vaccinated population respectively (p-value = 0.03).

**Table 1. Baseline characteristics of patients with COVID-19 survivors on follow-up (n = 481).**

| Characteristics | % (n) |
|---|---|
| **Age in years (mean ± SD)** | 56.9±14.3 |
| Age range (years) | 22–94 |
| **Sex** | |
| Male | 61.7% (297) |
| Female | 38.3% (184) |
| **Comorbidities** | |
| Diabetes | 55% (213) |
| Hypertension | 26% (101) |
| Other comorbidities | 19% (73) |
| **Vaccination status, n = 474** | |
| **Vaccinated (fully and partially vaccinated) n = 181.** | |
| Fully vaccinated | 19% (90) |
| Partially vaccinated | 19.2% (91) |
| Unvaccinated | 61.8% (293) |
| Vaccination status not known | 1.4% (7) |
| **Disease severity** | |
| **Non-Severe** | 35.1% (169) |
| **Severe/Critical** | 64.2% (312) |
| **Severe disease requiring NIV or IMV(Critical)** | 25.6% (120) |
| **Invasive mechanical ventilation (IMV)** | 3.3% (16) |
| **Non-invasive mechanical ventilation (NIV)** | 22.9% (110) |
| **Length of stay (index visit) days (median, IQR)** | 4(2–7) |
| **Readmission/revisit to the Emergency department** | 18.9% (91) |
| **Readmission requiring Inpatient hospitalization.** | 13.9% (67) |
| **Mortality rate on follow-up after the initial/index admission** | 4.58% (22) |

**Table 2. Reasons for readmission of COVID-19 survivors on follow-up (n = 91).**

| Reasons for readmission | % (n) |
|---|---|
| Respiratory impairment | 31.8% (29) |
| Infections | 17.6% (16) |
| Cardiac impairment (myocardial infarction, arrhythmias, heart failure) | 12% (11) |
| GI impairment | 17.6% (16) |
| Neurological | 2.2% (2) |
| Miscellaneous causes | 18.6% (17) |
| Respiratory failure | 27.4% (25) |
| Infections (UTI, Dengue, Pneumonia) | 17.1% (17) |
| Myocardial Infarction | 8.7% (8) |
| Acute gastroenteritis | 7.6% (7) |
| Venous thromboembolism (Pulmonary embolism, deep venous thrombosis) | 7.6% (7) |
| Gastrointestinal bleeding | 5.4% (5) |

Overall, 21.5% (n = 63) of the unvaccinated individuals required readmission to emergency/inpatient services, compared to the fully/partially vaccinated (13.8%, n = 25) group with a significant p-value. Twenty-four percent (n = 71) of the unvaccinated patients' admissions were secondary to pulmonary complications. New/worsened/persistent symptoms related to long COVID were more common in 37.9% (n = 105) of the unvaccinated as compared to 20% (n = 35) of the vaccinated population. Severe fatigue (33.9% vs. 17.1%), shortness of breath (30% vs. 17.1%), cough (20.9% vs. 8%), difficulty ambulating due to breathlessness (15.8% vs. 8%), and feverish feeling (8.3% vs. 2.2%) were also more common in the unvaccinated cohort than in the vaccinated cohort. More (45.7%, n = 80) patients were able to return to work in the vaccinated cohort, see **Tables 3 and 4**.

## Health-related characteristics of COVID-19 survivors stratified by disease severity

Approximately 459 survivors were interviewed for the presence of symptoms and other health-related ailments. The majority (65%, n = 315) of patients were asymptomatic at the time of the interview, with a higher percentage in the non-severe cohort than in the severe cohort (75.7% vs. 59.9%), p-value of 0.000. New/worsened/persistent symptoms associated with long COVID were present in 29.9% (n = 144) of the respondents, and these symptoms were more prevalent in the severe/critical cohort (35.5%, n = 111). The most prominent symptoms that persisted even one year after the discharge from the hospital, were severe fatigue (26.2%, n = 126) and shortness of breath/chest tightness/wheezing (24.1%, n = 116), followed by cough (15.2%, n = 73). Approximately 12.3% (n = 59) of the respondents had difficulty in ambulating due to shortness of breath, and many had more than one symptom. Compared to the non-severe group, a significantly higher number of individuals with severe/critical disease suffered from fatigue (15.3% vs. 32%), shortness of breath (13.6% vs. 29.8%), cough (8.8% vs. 18.5%), and difficulty in ambulating due to breathlessness (5.9% vs. 15.7%), p-value <0.005. Approximately 18.1% (n = 87), with 27.2% (n = 85) of the severe/critical cohort continued to require oxygen supplementation at home for a few weeks to months after discharge from the hospital. None of the patients was on supplemental oxygen at the time of the interview.

Among the 266 patients who answered questions regarding employment status, 45% (n = 207), as opposed to 12.8% (n = 59), were able to return to employment, with 40.9% (n = 188) able to return to work by 60 days after discharge. Even a year after discharge from the hospital, over 33.9% (n = 163) and 31% (n = 149) of patients reported emotional and

**Table 3. COVID-19 survivors' health-related characteristics stratified according to vaccination status.**

| Characteristics of disease on index admission based on Vaccination status, n = 474 | | | | |
|---|---|---|---|---|
| | **Vaccinated n = 181 (full/partial)** | **Unvaccinated n = 293** | **p-value** | **Crude Odds Ratio (95% CI)** |
| **Disease Severity** | | | | |
| Non-Severe | 41.5% (75) | 31.8% (93) | 0.032 | 1.52 (1.03–2.23) |
| Severe/Critical | 58.5% (106) | 68.2% (200) | | |
| Invasive mechanical ventilation | 1.10% (2) | 4.44% (13) | 0.063 | 4.15 (0.92–18.6) |
| Readmission | 13.8% (25) | 21.51% (63) | 0.005 | 1.81 (1.09–3.01) |
| Pulmonary complications$ | 15.47% (28) | 24.2% (71) | 0.023 | 1.74 (1.07–2.83) |
| Health-related characteristics on follow-up stratified on vaccination, n = 452 | | | | |
| | **Vaccinated (full/partial) = 175** | **Unvaccinated n = 277** | **p-value** | |
| Asymptomatic | 80 (140) | 62 (172) | 0.000 | 1.79 (1.31–2.45) |
| New/worsened/persistent symptoms related to long COVID | 20 (35) | 37.9 (105) | 0.000 | |
| Severe fatigue (n = 124) | 17.1 (30) | 33.9 (94) | 0.000 | 2.51 (1.57–3.9) |
| Minimal (n = 42) | 10.2 (18) | 8.6 (24) | | |
| Shortness of breath (SOB)/ chest tightness/ wheezing | 17.1 (30) | 30 (84) | 0.002 | 2.12 (1.33–3.40) |
| Cough (n = 72) | 8 (14) | 20.9 (58) | 0.000 | 3.04 (1.64–5.65) |
| Difficulty ambulating due to breathlessness | 8 (14) | 15.8 (44) | 0.015 | 2.30 (1.16 4.15) |
| Feverish | 2.2 (4) | 8.3 (23) | 0.013 | 3.89 (1.323–11.462) |
| Continued loss of taste and/or smell | 1.7 (3) | 2.8 (8) | 0.430 | 1.7 (0.44–6.5) |

$Pulmonary Complications (Respiratory impairment during readmission and/or on outpatient follow-up

~ Patients who had answered the relevant question included.

financial difficulties, respectively. Considering disease severity, the emotional impact was greater in the severe/critical disease group (p = 0.008). There were 6.1% (n = 28) patients, who had HADS anxiety and depression score > 11, with 3.4% (n = 16) and 4.3% (n = 20) of the patients having anxiety and depression respectively, but there was no significant difference observed between the two cohorts stratified on severity basis, see **Table 5**.

Most of the patients had HADS anxiety and depression score between 0–8 and the median HADS anxiety and depression score among the studied population was 2 (0–5), see **Table 6**.

After one year of infection, the majority (59.3%, n = 272) of patients continued to have NYHA class 1. However, 12% (n = 55) and 2.6% (n = 13) of the patients became NYHA 4, compared to 7% (n = 32), and 0.20% (n = 1) previously, see **Fig 2**.

**Table 4. COVID-19 survivors' return to work and HADS score stratified according to vaccination status.**

| | **Vaccinated (full/partial) n = 175** | **Unvaccinated n = 277** | **p-value** |
|---|---|---|---|
| Return to Employment (n = 266, NA = 193) | | | 0.000 |
| Yes (202) | 45.7 (80) | 44 (122) | |
| No (61) | 4 (7) | 19.4 (54) | |
| Able to return to work in 60 days following discharge. | | | 0.072 |
| Yes | 42.2 (74) | 39.7 (110) | |
| No | 3.4 (6) | 7.5 (21) | |
| Anxiety (HADS-A score >11) | | | |
| n = 16 | 5.7 (10) | 2.1 (6) | 0.047 |
| Depression (HADS-D score >11) | | | |
| n = 20 | 2.8 (5) | 5.4 (15) | 0.198 |
| Combined anxiety and depression (HADS-A | | | |
| and/or HADS-D score >11) (28) | 5.7 (10) | 6.4 (18) | 0.73 |

Table 5. COVID-19 survivor's health-related characteristics stratified on severity basis (n = 459).

| Follow-up data of study participants | Total | Non-Severe %, n = 169 | Severe/Critical | p-value |
|---|---|---|---|---|
| | %, n = 459 | | %, n = 312 | |
| Asymptomatic | 65.5% (315) | 75.7 (128) | 59.9 (187) | 0.000 |
| New/worsened/persistent symptoms related to illness (Long COVID) | 29.9% (144) | 19.5 (33) | 35.5 (111) | 0.000 |
| Severe fatigue | 26.2% (126) | 15.3 (26) | 32 (100) | 0.000 |
| Minimal fatigue | 9.3% (43) | 5.9 (10) | 10.5 (33) | |
| Shortness of breath (SOB)/ chest tightness/ wheezing | 24.1% (116) | 13.6 (23) | 29.8 (93) | 0.000 |
| Cough | 15.2% (73) | 8.8 (15) | 18.58 (58) | 0.005 |
| Difficulty ambulating due to breathlessness. | 12.3% (59) | 5.9 (10) | 15.7 (49) | 0.001 |
| Feverish | 5.6% (27) | 3.5 (6) | 6.7 (21) | 0.151 |
| Continued loss of taste and/or smell | 2.3% (11) | 1.18 (2) | 2.8 (9) | 0.343 |
| Supplemental oxygen | 18.1% (87) | 0.5 (1) | 27.2 (85) | 0.000 |
| Return to employment. | | | | |
| Yes | 45% (207) | 46.7 (79) | 41 (128) | 0.165 |
| No | 12.8% (59) | | | |
| Not Applicable | 42% (193) | | | |
| Able to return to work within 60 days following discharge | 40.9% (188) | 46.1 (78) | 35.2 (110) | 0.001 |
| Emotional impact | 33.9% (163) | 43.7 (74) | 28.5 (89) | 0.008 |
| Financial impact | 31% (149) | 36.6 (62) | 27.8 (87) | 0.208 |
| Anxiety (HADS-A score >11) | 3.4% (16) | 3.5 (6) | 3.2 (10) | 0.83 |
| Depression (HADS-D score >11) | 4.3% (20) | 4.14 (7) | 4.1 (13) | 0.994 |
| Combined anxiety and depression (HADS-A and/or HADS-D score >11) | 6.1% (28) | 5.3 (9) | 6 (19) | 0.737 |

## Multivariable analysis for factors associated with long COVID

On multivariable analysis, after adjusting for age, gender, presence of co-morbid conditions, and disease severity, lack of vaccination was found to be an independent predictor of long COVID with an Odds ratio of 2.42 (95% CI 1.52–3.84). Fully vaccinated and partially vaccinated patients had 62% and 56% reduced risk of developing long COVID respectively, see **Table 7**.

## Discussion

There is increasing anecdotal awareness of patients with "Long COVID" in whom residual symptoms persist beyond the acute viral illness [17–20]. This study reports the long-term health outcomes of COVID-19 survivors at 1 year following hospital discharge in a large cohort of patients. Our study observed that the patients who required hospitalization due to severe/critical COVID-19 infection or had an unvaccinated status continued to have debilitating symptoms and functional status with significant financial and emotional impact on their lives. Additionally, it was observed that patients with severe/critical disease and unvaccinated status had higher readmission rates due to various reasons (unrelated to long COVID).

Table 6. Anxiety and depression in COVID-19 survivors.

| Anxiety | | Depression | |
|---|---|---|---|
| HADS score | Total % (n) | HADS score | Total % (n) |
| Category 1: 0–7 | 91.7 (421) | Category 1: 0–7 | 89.5 (411) |
| Category 2: 8–10 | 4.8 (22) | Category 2: 8–10 | 6.1 (28) |
| Category 3: > = 11 | 3.5 (16) | Category 3: > = 11 | 4.4 (20) |
| HADS Anxiety score (median IQR) | 2 (0–4) | HADS Depression score (median IQR) | 2 (0–5) |

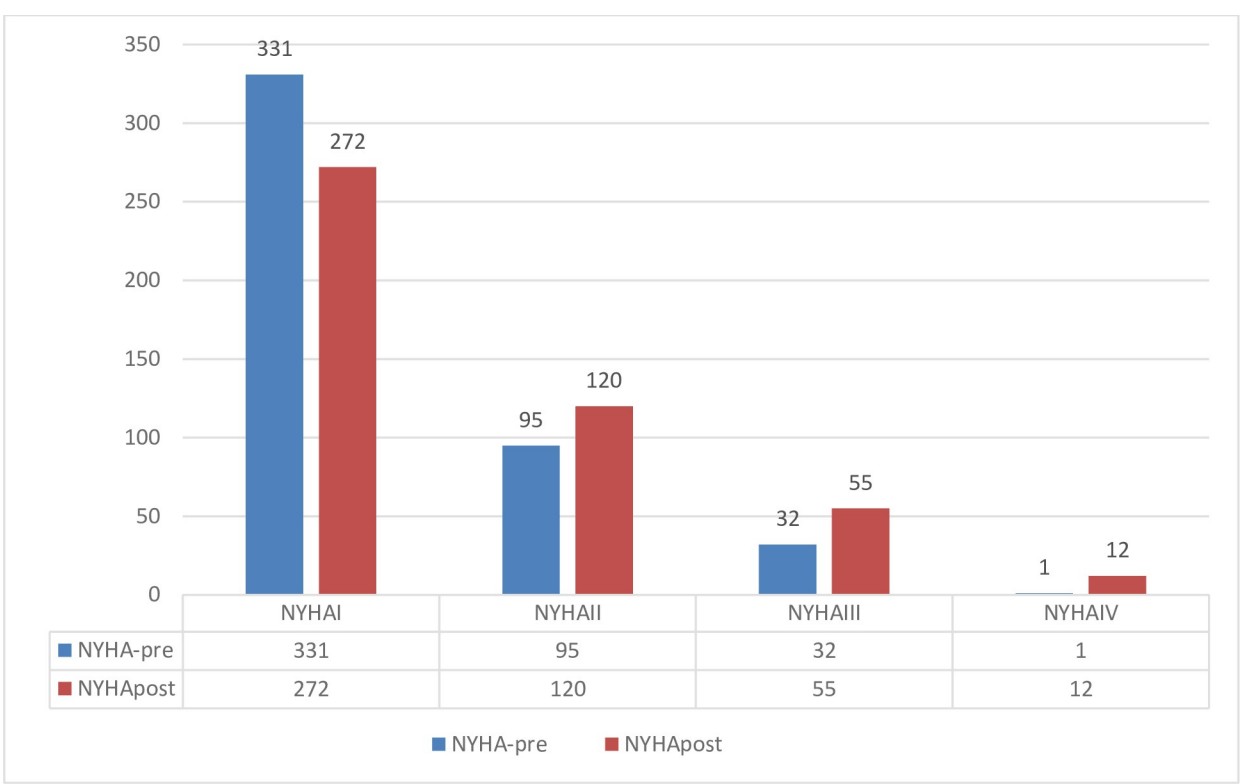

**Fig 2. NYHA class pre- and post-hospitalization at one-year follow-up (n = 459).**

Xue Zhang et al. and MM Maestre-Muñiz et al. did a similar study and reported the persistence of symptoms in COVID-19 survivors one year after discharge from the hospital in 45% and 56.9% of the interviewees respectively. Fatigue, breathlessness, sweating, chest tightness, anxiety, ageusia, anosmia, and myalgia were the most common symptoms. Fatigue was higher in patients with older age, female sex, and severe disease [21, 22]. Compared to this, most (65.5%) of our patients were asymptomatic and only 29.9% reported persistent symptoms related to the illness; however, fatigue, breathlessness, and chest tightness continued to be common among both populations. MM Maestre-Muñiz and colleagues conducted a thorough study, but they did not categorize symptoms based on disease severity and vaccination status. On the other hand, Xue Zhang and their team stratified the cohort into non-severe and severe disease categories, but the criteria for defining COVID-19 severity were not clearly outlined.

**Table 7. Multivariate analysis for factors associated with long COVID.**

| Variables | adjusted Odds ratio (95% Confidence interval) | p-value |
|---|---|---|
| **Female gender** | 1.14 (0.76–1.70) | 0.654 |
| **Age > 60 years** | 2.62 (1.70–4.04) | 0.000 |
| **>2 comorbid conditions** | 1.24 (0.70–2.21) | 0.463 |
| **Severe Disease** | 1.84 (1.13–2.97) | 0.014 |
| **Fully vaccinated** | 0.38 (0.20–0.7) | 0.002 |
| **Partially vaccinated** | 0.44 (0.24–0.80) | 0.007 |

aOR(adjusted odd ratio) for unvaccinated 2.42 (1.52–3.84)

Additionally, neither of these studies assessed the emotional and financial hardships experienced by COVID-19 survivors. Lixue Huang et al. conducted a study and observed that around 49% of the patients had at least one sequela at 12 months follow-up. Dyspnea and depression were reported in 30% and 26% of the patients compared to 24.1% and 4.3% respectively in our study. This difference in observations made for depression and anxiety may be due to the difference in the scales. Lixue Huang et al. study is remarkable in a way as it included face-to-face interviews, however, stratification according to the severity of the disease and vaccination status remains to be addressed [23]. We also found two regional studies by FNU Shivani et al. (12 months follow-up, hospitalized patients) and Madeeha Khan et al. (12-week follow-up, only 16.7% hospitalized patients), and observed that fatigue and breathing problems were the most common symptoms of COVID-19 survivors [24, 25]. In both regional studies, further analysis based on the severity of the disease and vaccination status was not done.

A study conducted in an intensive care unit (on NIV/invasive ventilation/high nasal flow cannula), found that 82% of the patients suffered from fatigue (compared to 35% of our population). This study reported health impairments in critically ill patients but had limitations due to the absence of a control group and a small sample size, including patients from the first wave [26].

Taquet et al. analysed electronic healthcare data from 81 million patients and reported that 36.5% of the COVID survivors developed long COVID symptoms between 3 and 6 months [27]. Another study conducted in Germany between October 2020 and August 2021, on 51,630 patients observed that 8.3% of the patients in general practices suffered from long COVID i.e., in the initial pandemic waves [28]. A meta-analysis of 137 studies revealed that the initial symptoms decreased from 92% to 55% at 1 month and remained stable at 54% at 6 and 12 months [29]. Another meta-analysis showed no significant difference in the short-term (1-month) and long-term (>6 months) prevalence of post-acute sequelae of COVID-19 (PASC) among patients requiring hospitalization. Almost 54% of patients reported at least one PASC at 6 or more months. Notably, 79% of the studies included in the analysis were from high-income counties and they did not include data regarding symptom reduction, improvement, or deterioration [30].

In a systematic review of the effect of vaccination in reducing long COVID symptoms, Byambasuren et al. reported the lowest odds ratio of 0.48 to 1.01 for developing long COVID even with "any dose" of vaccine before infection [31]. Most studies have reported a reduction in long COVID risk with immunization, ranging between 15% to 41% [10]; however, few studies have been reported on symptom-specific reduction. Taquet et al. in a study on breakthrough infections in 10,024 COVID-19 patients, reported lower odds of death (0.66) in vaccinated patients. Furthermore, receiving two vaccine doses i.e., being fully vaccinated was associated with lower risk for most of the long COVID symptoms but not all Post-COVID Conditions (PCC) [32]. Similarly, Kuodi et al. from Israel observed that patients vaccinated twice with the BNT162b2 vaccine reported fewer PCC symptoms [33]. From the UK, at 12 weeks of follow-up, Daniel et al. [34] observed that full vaccination was associated with a 41% reduction in the odds of developing long COVID. In another cohort-based study from the UK, Antonelli et al. reported a risk reduction halved in vaccinated patients at a four-week follow-up [35]. Similar to their findings, our study also noted a decrease in post-COVID symptoms with vaccination, especially among patients under 60 years. Mizrahi et al. reported a higher risk of dyspnoea in unvaccinated with a hazard ratio of 1.58 which was similar to OR 2.42 in our study [36].

Due to long COVID, an estimated 9–40% of patients requiring hospitalization were absent from work at 60 and 90 days after discharge [37]. Around 86–88% of the patients who were employed before COVID-19 had returned to their original work at 12 months, and all the

patients with ARDS achieved independence in ADL at 12 months [23, 38]. Karpman et al reported that [39], adult patients with long COVID (PCC) faced more barriers to accessing health care due to costs; lack of specific clinics, and finding appointments. Moreover, the economic implications of the long COVID pandemic have not been studied in LMIC, where healthcare resources are already constrained.

Based on our study and literature review we have certain recommendations that might result in improvement in data collection on long COVID and improve preparation for the post-Pandemic wave of post-COVID conditions [40]. We recommend long-term studies across 1–3 years to estimate the true number of cases developing long COVID, a few examples of these are a LOCUS study (Long COVID–Understanding Symptoms, events and Use of Services in Portugal) [41] and an open registry with 6 monthly follow-up for 3 years in Bavaria [42]. Using data on vaccination available in government databases; the long-term impact of the total number of doses; the need for boosters and the time to vaccination in our population can be evaluated. To unveil the true burden of disease, data collection in terms of DALYs should be conducted in comparison to age-standardized mortality rates, incidence, and prevalence, as this will also be useful in terms of comparison of disease burden with other NCDs (Non-Communicable diseases) across different countries. In, Germany, for 2020, approximately 1-year DALYs were found to be 305,641 [43]. Estimates from Scotland ranged from 96,500 to 108,200 and surprisingly COVID-19 DALYs were second to ischemic heart disease among NCDs [44]. Importantly, the mortality rate contributed a small share to this morbidity in DALYs, and therefore allocation of resources should be prioritized not only for the prevention of mortality but also for the reduction in long COVID [45, 46]. Furthermore, we also recommend symptom-specific data collection, registries, multidisciplinary research clinics, and the development of clinical practice guidelines. Training of healthcare professionals along with improving access to specialist care for conditions such as post-COVID fibrosis, is paramount in reducing costs and improving affordability.

Our study has certain limitations. This single-centre observational study was conducted in a large private tertiary care hospital; without control groups, therefore, the results cannot be generalized to the entire population. Second, 44% of the eligible population were not interviewed, because they were not accessible, and few declined to participate in the study. There was also survivor bias as we were not able to recruit the patients who died after a few months of discharge. Another constraint of this study was the potential limitation associated with telephone interviews, which may not be as precise as face-to-face interactions. Furthermore, the interviews relied on subjective information provided by the participants and did not incorporate objective evidence obtained through further investigative procedures. Readmission frequency may have been underreported as few patients may have been readmitted to another healthcare facility. COVID-19 is known to unmask preexisting diabetes and cause thyroid disease, the symptoms of which may resemble those of long COVID, we did not assess this in our study as these diagnoses (new-onset diabetes or hypothyroidism) would have required a review of the diagnostic labs/criteria.

Despite these limitations, to our knowledge, this is the first reported data from Pakistan, on the outcomes of patients past one year of hospitalization with COVID-19 infection. This study not only evaluated the impact of severity of the disease on long COVID but also assessed the role of vaccination status in prevention. In comparison to other studies, we did not rely solely on electronic health records but also interviewed patients for symptom evaluation. The emotional, mental, and financial toll of the disease was also evaluated and risk factors specifically predictive of long Covid were identified in our population. Another strength of the study is the inclusion of patients with positive RT-PCR or rapid antigen tests as other studies included patients based on clinical symptoms without confirmatory tests.

## Conclusion

This study evaluated the long-term consequences of COVID-19 on symptoms, mental health, and functional recovery according to disease severity a year after hospitalization. Based on our findings, individuals who experienced severe and critical forms of the disease are more susceptible to enduring debilitating symptoms like fatigue and breathlessness, persisting for several months following their hospital discharge. Patients with severe and critical disease also reported financial and emotional difficulties compared to those with non-severe disease. Thus, healthcare providers should emphasize the rehabilitation of COVID-19 survivors along with their long-term follow-up with necessary investigations and treatment. Vaccination resulted in a reduction in both mortality and risk of long COVID, this finding can be used to emphasize the long-term importance of vaccination after the pandemic and might increase vaccination uptake rates.

## Supporting information

**S1 Data. Additional data that provides an overview of the health-related characteristics of COVID-19 survivors based on their vaccination status (i.e., vaccinated, partially vaccinated, and unvaccinated) can be found in S1 and S2 Tables in S2 File.**
(CSV)

**S1 File.**
(DOCX)

**S2 File. Contains supporting tables.**
(DOCX)

## Author Contributions

**Conceptualization:** Madiha Ismail.

**Data curation:** Taymmia Ejaz, Zarnain Shah, Mohammad Shahzaib, Hassan Masood Jafri.

**Formal analysis:** Samar Fatima, Madiha Ismail, Taymmia Ejaz.

**Investigation:** Samar Fatima, Zarnain Shah, Summaya Fatima, Hassan Masood Jafri.

**Methodology:** Samar Fatima, Madiha Ismail.

**Project administration:** Samar Fatima, Zarnain Shah, Summaya Fatima, Mohammad Shahzaib, Hassan Masood Jafri.

**Resources:** Summaya Fatima.

**Software:** Taymmia Ejaz, Summaya Fatima, Mohammad Shahzaib.

**Supervision:** Samar Fatima, Madiha Ismail, Zarnain Shah.

**Validation:** Samar Fatima, Madiha Ismail, Taymmia Ejaz, Mohammad Shahzaib.

**Visualization:** Madiha Ismail, Taymmia Ejaz, Mohammad Shahzaib.

**Writing – original draft:** Samar Fatima, Madiha Ismail, Mohammad Shahzaib, Hassan Masood Jafri.

**Writing – review & editing:** Samar Fatima, Madiha Ismail, Taymmia Ejaz.

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
