## [Decision Letter · Decision Letter 0]

21 Aug 2023

PONE-D-23-22756The Post-Pandemic Aftermath in a LMIC: A 12-month follow-up study on Long COVID symptoms and the impact of vaccination in hospitalized patientsPLOS ONE

Dear Dr. Ismail,

Thank you for submitting your manuscript to PLOS ONE. After careful consideration, we feel that it has merit but does not fully meet PLOS ONE’s publication criteria as it currently stands. Therefore, we invite you to submit a revised version of the manuscript that addresses the points raised during the review process.

Long covid is a topic of interest in recent times and can be studied in relation to the variant, severity of disease, comorbid conditions and vaccination status 

The reviewers have suggested changes in the analysis and some minor changes which will help strengthen the manuscript.

We look forward to receiving your revised manuscript.

Kind regards,

Yatin N. Dholakia, MD

Academic Editor

PLOS ONE

Journal Requirements:

Additional Editor Comments:

Long Covid has been the topic of interest in recent times. The symptoms depend on various factors such as vaccination status, age, comorbidities and severity of SARS Cov2 infection and many other which are under research.

The manuscript has been reviewed and there are comments from the reviewers which will help strengthen the same.

Reviewers' comments:

Reviewer's Responses to Questions

**Comments to the Author**

1. Is the manuscript technically sound, and do the data support the conclusions?

Reviewer #1: Yes

Reviewer #2: Partly

2. Has the statistical analysis been performed appropriately and rigorously? 

Reviewer #1: Yes

Reviewer #2: Yes

3. Have the authors made all data underlying the findings in their manuscript fully available?

Reviewer #1: Yes

Reviewer #2: Yes

4. Is the manuscript presented in an intelligible fashion and written in standard English?

Reviewer #1: Yes

Reviewer #2: Yes

5. Review Comments to the Author

Reviewer #1: 1. Severe/critical disease was present in 64% of the population, and 25.6% required non-invasive/invasive mechanical ventilation.

64% of the 481 patients analysed, right ? Please write numbers along with percentage

2. The authors mention that ,”On Long COVID, to the best of our, knowledge, prevalence data, estimates of the at-risk population, and the burden have not been reported from Pakistan”. However there is a study , “Analysing the psychosocial and health impacts of Long COVID in Pakistan: A cross sectional study” from AGKU by Madeeha Khan et al. Please clarify this and highlight the differences between the two studies.

4. As per the above study (by Madeeha Khan et al), A small proportion (n=9; 16.7%) of these long haulers were hospitalized during the acute phase. In other studies too, symptoms of long COVID have been observed in patients who had mild symptoms, in contrast to your study where most had severe symptoms.

5.Expedited NOT expeditated

6. Low antibodies titres in patients with COVID-19 have been reported to be predictive of Long COVID regardless of hospitalization status[10], and Perlis et al from USA reported that full vaccination resulted in a lower risk for Long COVID with OR 0.72[11]

Suggest rephrasing to ..Low antibody titres to SARSCoV 2 in patients………….

7. The abstract mentions the study period as between 1st January 21 to 30th June 21, whereas the article mentions it as 1st Feb 21 to 7th June 21.

8. COVID 19 is known to unmask pre existing diabetes and also cause thyroid disease, the symptoms of which may resemble those of Long COVID. Were patients asked for new onset diabetes or hypothyroidism, following COVID 19 ?

9. Grammatical errors need to be addressed.

Reviewer #2: Dr Fatima and colleagues carried out a prospective study to explore the association between vaccination status and long COVID symptoms after hospitalization for COVID-19 in Pakistan. The following comments may be useful to the authors.

General comments:

1. Page 1, Title, “The Post-Pandemic Aftermath in a LMIC: A 12-month follow-up study on Long COVID symptoms and the impact of vaccination in hospitalized patients”: This title seems vague. The authors may consider rephrasing to “Association between vaccination status and occurrence of long COVID symptoms after hospitalization for COVID-19 in a low- or middle-income country”.

2. In accordance with my first comment, the authors may wish to reconstruct their manuscript focusing on the association between vaccination status and long COVID. Although there are studies on this subject from high-income countries (summarized in Byambasuren O et al. BMJMed 2023), data from low- or middle-income countries (LMIC) are scarce and useful.

3. If the authors choose to follow the above suggestion, then they should reformat their Tables. For example, Table 1 and Table 3 could present baseline characteristics and long COVID symptoms of COVID-19 survivors according to vaccination status (i.e., non-vaccinated, partially vaccinated and fully vaccinated).

4. Following the above comment, the categorization into three groups (i.e., non-vaccinated, partially vaccinated and fully vaccinated) may be more informative than the current categorization (Tables 5-6) into two groups (i.e., non-vaccinated versus partially/fully vaccinated).

5. This categorization into three groups could be applied in the multivariate analysis exploring factors associated with long COVID (Table 6 of page 19). It would be nice to see whether there is a “dose-dependent” effect; i.e., is the odds ratio for the association between full vaccination status and long COVID nominally smaller that the odds ratio for the association between partial vaccination status and long COVID?

6. In general, language editing may substantially benefit the manuscript. In specific, the authors may wish to use short sentences. Take for example, in page 2, Abstract, the long sentence: “On Long COVID, to the best of our, knowledge, prevalence data, estimates of the at-risk population, and the burden have not been reported from Pakistan, due to absence of prospective longitudinal studies on COVID-19 patients”.

Specific comments:

7. Page 3, Abstract: In the Results paragraph, it is written that “The most prominent symptoms that persisted even one year after infection…”. However, in the Conclusions paragraph, it is written that “This study reports the long-term health outcomes of COVID-19 survivors at 1-year following hospital discharge…”. The authors may wish to clarify when they assessed the study outcome; at one year after infection or at one year after hospital discharge?

8. Page 6, Eligibility criteria and data collection, “Adult patients admitted between February 1, 2021, and June 7, 2021…”: In page 2, Abstract, it was written “…recruited patients aged > 18 years who were admitted between 1st January and June 30, 2021”. The authors may wish to correct such inconsistencies throughout their manuscript.

9. There are two Tables 6; one in page 18 and one in page 19.

10. Page 19, Figure 2 could be polished. For example, the authors could delete the “Chart title”.

11. Pages 20-27: Discussion could be substantially shortened.

6. PLOS authors have the option to publish the peer review history of their article (what does this mean?). If published, this will include your full peer review and any attached files.

Reviewer #1: No

Reviewer #2: No

---

## [Author Response · Author response to Decision Letter 0]

1 Oct 2023

Response to Reviewer

We would like to express our sincere gratitude for considering our manuscript, titled "The Post-Pandemic Aftermath in a LMIC: A 12-month follow-up study on Long COVID symptoms and the impact of vaccination in hospitalized patients," for review at PLOS ONE.We greatly appreciate the time and effort put forth by both you and the reviewers in evaluating our work. In response to the insightful comments provided by the reviewers, we have diligently revised and modified our manuscript. Every suggestion made by the reviewers has been incorporated into the manuscript to enhance its quality and clarity.

With these revisions, we believe that our manuscript is now well-aligned with the high standards of PLOS ONE and is suitable for publication. We eagerly await your feedback on the revised submission and are ready to address any further questions or comments you may have.

Yours sincerely,

On behalf of all the authors,

Dr.Madiha Ismail

Journal Requirements:

Reply:

●We have made corrections according to PLOS ONE's style requirementsand also copyedit the manuscript for language usage, spelling, and grammar.

●The name of the colleague that has edited our manuscript is Akbar Shoukat Ali (Research Associate, Department of Medicine, Aga Khan University Hospital, Karachi, Pakistan)

●A copy of the revised manuscript showing track changes has been uploaded as a *supporting information* file)

● A clean copy of the edited manuscript has been uploaded as the new *manuscript* file

"The following represents our response to the reviewers' comments."

Reviewer 1

Comment 1:

Severe/critical disease was present in 64% of the population, and 25.6% required non-invasive/invasive mechanical ventilation.

64% of the 481 patients analysed, right? Please write numbers along with percentage

Reply:

We have added numbers with the percentages as per your recommendation. This has also been corrected and highlighted in the manuscript. For further clarification, we have also added here.

Severe/critical disease was present in 64%(312/481) of the population, and 25.6%(120/481) required non-invasive/invasive mechanical ventilation

Comment 2:

The authors mention that ,”On Long COVID, to the best of our, knowledge, prevalence data, estimates of the at-risk population, and the burden have not been reported from Pakistan”. However there is a study ,“Analysing the psychosocial and health impacts of Long COVID in Pakistan: A cross sectional study” from AGKU by Madeeha Khan et al. Please clarify this and highlight the differences between the two studies.

Reply :

Thank you for bringing this study to our attention. We did not include this study in our manuscript because this is a pre-print version. Additionally, at the time of manuscript preparation, this preprint was notavailable and it was published online on May 23, 2023.The participants included in the study done byMadeeha Khan et al. differ from ours and we are mentioning the differenceshere and alsohaveincluded this in the discussion section of the manuscript.(doi: https://doi.org/10.1101/2023.05.22.23290323)

1) In our study we looked for Long COVID symptoms in patients who were hospitalized between February 1, 2021, and June 7, 2021, and interviewed them for Long COVID one year after their discharge from the hospital, while in the study by Madeeha Khan et al.,only 8.4% (n=13)required hospitalizationand most of participant included were outpatient.Moreover, only a small proportion (n=9; 16.7%) of these long haulers were hospitalized during the acute phase.

2) The author evaluated Long COVID symptoms at 12 weeks following infection from COVID-19 disease, as compared to our study in which we interviewed the patients one year after their discharge from the hospital.

3) Madeeha Khan et al. assessed the health symptoms as wellas psychological impact but did notanalyze it based onthe severity of the disease and vaccination status.

4) Among the 155 COVID-19-positive patients, 54 (35%) continued to experience symptoms for more than 12 weeks after infection, compared to 29.9% of our patients at the one-year mark.

5) The most frequent symptoms in the Madeeha et al study were muscle problems and fatigue (14.7%) followed by breathing difficulties such as breathlessness, dyspnea, painful breathing, or cough (12.6%). However, in our study, this percentage was higherwith fatigue reported at 26.2% and symptoms like shortness of breath, chest tightness, and wheezing at 24.1%.The elevated occurrence of this disparity within our patient population may stem from the fact that a substantial majority of our patients, comprising 64% (312 out of 481), suffered from severe or critical disease. Moreover, all our patients were hospitalized.

6) The author used different scales for assessing stress, anxiety and depression as compared to our study. 

7) We only included patientswith confirmed positiveSARS-CoV-2 RT-PCR. However, in their study, among 300 patients, 155 (51.7%) had COVID-19 either confirmed by positive RT-PCR and/or diagnosed by a general physician on clinical grounds without any testing.

Comment 3: 

As per the above study (by Madeeha Khan et al.), A small proportion (n=9; 16.7%) of these long haulers were hospitalized during the acute phase. In other studies too, symptoms of long COVID have been observed in patients who had mild symptoms, in contrast to your study where most had severe symptoms.

Reply :

In our study as well, patients with non-severe COVID suffered from long COVID symptoms, however as compared to severe/critical disease, individuals with severe/critical disease were more likely to have long COVID. This observation aligns with the findings of Xue Zhang et al. (https://www.ncbi.nlm.nih.gov/pmc/articles/PMC8482055/). 

Furthermore, weonly included patients necessitating hospitalization and testing positive through RT-PCR or Antigen tests. Given that patients with mild disease typically do not require hospitalization, our study naturally demonstrated a lower proportion of such cases in comparison.Madeeha Khan et al reportedthat only 13 (8.4%) in their study population hadrequired hospitalization.

Comment 4: 

Expedited NOT expeditated

Reply :

We have corrected the spelling error.

Comment 5:

Low antibody titres in patients with COVID-19 have been reported to be predictive of Long COVID regardless of hospitalization status[10], and Perlis et al from USA reported that full vaccination resulted in a lower risk for Long COVID with OR 0.72[11]

Suggest rephrasing to ..Low antibody titres to SARSCoV 2 in patients………….

Reply :

This has been rephrased to:

It has been observed that low antibody titres to SARS CoV 2 have been associated with a greater likelihood of experiencing Long COVID, regardless of hospitalization status. Perlis et al. conducted a study and observedthat full vaccination resulted in a lower risk of developing long COVID with an OR of 0.72. 

Comment 6:

The abstract mentions the study period as between 1st January 21 to 30th June 21, whereas the article mentions it as 1st Feb 21 to 7th June 21.

Reply :

This has been corrected and highlighted in the manuscript. The studyrecruited patients who were admitted between February 1, 2021, and June 7, 2021, and it has been rechecked from the data.

Comment 7:

COVID-19 is known to unmask pre-existing diabetes and also cause thyroid disease, the symptoms of which may resemble those of Long COVID. Were patients asked for new onset diabetes or hypothyroidism, following COVID 19 ?

Reply :

Thank you for bringing this to our attention. It is an area of interest for us too. However, these could not be assessed in our study as these diagnoses(new onset diabetes or hypothyroidism) would have required a review of the diagnostic labs/criteria. We will mention this in our limitation section.

Comment 8:

Grammatical errors need to be addressed.

Reply :

We have corrected the grammatical errors and the manuscript has been revised.

Reviewer 2

Comment 1:

Page 1, Title, “The Post-Pandemic Aftermath in a LMIC: A 12-month follow-up study on Long COVID symptoms and the impact of vaccination in hospitalized patients”: This title seems vague. The authors may consider rephrasing to “Association between vaccination status and occurrence of long COVID symptoms after hospitalization for COVID-19 in a low- or middle-income country”.

Reply :

We appreciate your input and agree that clarity in the title is important. We have considered your suggestion, and our revised title is now:

Association between Long COVID and Vaccination: A 12-Month Follow-up Study in a Low- to Middle-Income Country

Comment 2:

In accordance with my first comment, the authors may wish to reconstruct their manuscript focusing on the association between vaccination status and long COVID. Although there are studies on this subject from high-income countries (summarized in Byambasuren O et al. BMJMed 2023), data from low- or middle-income countries (LMIC) are scarce and useful.

Reply :

In response to the suggestion provided in your first comment, we appreciate the emphasis on exploring the association between vaccination status and long-term COVID. It is indeed a valuable point that data from low- or middle-income countries (LMIC) are limited in this context.On your advice, we have reconstructed our manuscript with more focus on the association between vaccination status and long COVID.

However, in our study, we aimed to address not only the association between long COVID and vaccination but also its relationship with the severity of the disease. This comprehensive approach allows us to provide a more holistic understanding of Long COVID in our specific population.

Comment 3:

If the authors choose to follow the above suggestion, then they should reformat their Tables. For example, Table 1 and Table 3 could present baseline characteristics and long COVID symptoms of COVID-19 survivors according to vaccination status (i.e., non-vaccinated, partially vaccinated and fully vaccinated).

Reply :

Thank you for your suggestion. Initially, we had intended to follow the same approach. However, due to a disparity in the total population number between the baseline characteristics (n=481) and the total number of patients for whom vaccination status was available (n=474, with Long COVID symptoms assessed in 452patient-survivors), we decided to create separate tables, one for baseline characteristics and another for vaccination. This choice was made to prevent any potential confusion during the reading of our manuscript as well as in the analysis. Additionally, we aimed to assess Long COVID based on its severity.

Furthermore, for chi-square testing and calculation of the odd ratio, we needed two categories, so we opted for vaccinated(partially/fully) and unvaccinated groups. 

However, recognizing the merit of your insightful suggestion, we have taken it into account and have included a comprehensive table detailing the disparities in outcomes among partially vaccinated, fully vaccinated, and unvaccinated individuals in the supplementary file.We are also including these tables in this document. 

Table 8: COVID-19 survivors’ health-related characteristics stratified according to vaccination status.

Characteristics of disease on index admission based on Vaccination status, n=474

 Fully vaccinated % (n=90) Partially vaccinated % (n=91) Unvaccinated % (n=293) p-value

Disease Severity 

Non-Severe 45.6 (41) 37.4 (34) 31.8 (93) 0.052

Severe/Critical 54.4 (49) 62.6 (57) 68.2 (200) 

Invasive mechanical ventilation 0 2.2 (2) 4.44 (13) 0.077

Readmission 15.6 (14) 12.2 (11) 21.51 (63) 0.056

Pulmonary complications$ 13.4 (12) 17.6 (16) 24.2 (71) 0.023

Health-related characteristics on follow-up stratified on vaccination, n=452

 Fully vaccinated n=87 Partially vaccinated

n= 88 Unvaccinated n=277 p-value

Asymptomatic 80.5 (70) 79.5 (70) 62.1 (172) <0.000

New/worsened/persistent symptoms related to Long COVID 19.5 (17) 20.5 (18) 37.9 (105) 0.000

Fatigue (n=124) 17.2 (15) 17.1 (15) 33.9 (94) 0.000

Minimal (n=42) 11.9 (5) 14.8 (13) 8.6 (24) 

Shortness of breath (SOB)/ chest tightness/ wheezing 14.9 (13) 19.1 (17) 30 (84) 0.005

Cough (n= 72) 3.4 (3) 12.5% (11) 20.9 (58) 0.000

Difficulty ambulating due to breathlessness 9.2 (8) 6.7 (6) 15.8 (44) 0.04

Feverish 1.1 (1) 3.4 (3) 8.3 (23) 0.025

Continued loss of taste and/or smell 2.3 (2) 1.1 (1) 2.8 (8) 0.83

Table 9: COVID-19 survivors’ return to work and HADS score stratified according to vaccination status.

 Fully vaccinated % (n=87) Partially vaccinated

% (n= 88) Unvaccinated % (n=277) p-value

Return to Employment (n=266, NA=193) n=40 n=47 n=176 

Yes (202) 87.5 (35) 95.7 (45) 69.4 (122) 0.000

No (61) 12.5 (5) 4.3 (2) 30.7 (54) 

Able to return to work in 60 days following discharge. n=36 n=44 n=131 

Yes 94.4 (34) 90.0 (40) 84 (110) 0.177

No 5.6 (2) 3.4 (6) 16 (21) 

Anxiety (HADS-A score >11) 

 n =16 8 (7) 3.4(3) 2.1(6) 0.035

Depression (HADS-D score >11) 

n =20 4.6 (4) 1.1 (1) 5.4 (15) 0.246

Combined anxiety and depression (HADS-A and/or HADS-D score >11) (28) 

 8 (7) 3.4 (3) 6.4 (18) 0.42

Comment 4:

Following the above comment, the categorization into three groups (i.e., non-vaccinated, partially vaccinated and fully vaccinated) may be more informative than the current categorization (Tables 5-6) into two groups (i.e., non-vaccinated versus partially/fully vaccinated).

Reply :

This table has been added as a supplementary file.

Comment 5:

This categorization into three groups could be applied in the multivariate analysis exploring factors associated with long COVID (Table 6 of page 19). It would be nice to see whether there is a “dose-dependent” effect; i.e., is the odds ratio for the association between full vaccination status and long COVID nominally smaller than the odds ratio for the association between partial vaccination status and long COVID?

Reply :

As per your advice, this has been changed.

On multivariable analysis, after adjusting for age, gender, presence of co-morbid conditions and disease severity, vaccination status was found to be an independent predictor of Long COVID with an Odds ratio of 2.42(95% CI 1.52-3.84) for the unvaccinated cohort. Fully vaccinated and partially vaccinated patients had 62% and 56% reduced risk of developing Long COVID respectively. See Table 7.

Table 7: Multivariate analysis for factors associated with Long COVID

Variables adjusted Odds ratio (95% Confidence interval) p-value

Female gender 1.14 (0.76-1.70) 0.654

Age > 60 years 2.62(1.70-4.04) 0.000

>2 comorbid conditions 1.24(0.70-2.21) 0.463

Severe Disease 1.84(1.13-2.97) 0.014

Fully vaccinated 0.38 (0.20-0.7) 0.002

Partially vaccinated 0.44(0.24-0.80) 0.007

aOR(adjusted odd ratio) for unvaccinated 2.42(1.52-3.84)

Comment 6:

In general, language editing may substantially benefit the manuscript. In specific, the authors may wish to use short sentences. Take for example, on page 2, Abstract, the long sentence: “On Long COVID, to the best of our, knowledge, prevalence data, estimates of the at-risk population, and the burden have not been reported from Pakistan, due to absence of prospective longitudinal studies on COVID-19 patients”.

Reply :

We appreciate your suggestion regarding language editing. Ensuring the manuscript's language and presentation meet the highest standards is crucial. We have worked on improving the manuscript's overall language and readability to enhance its clarity and impact

We have shortened the sentence and changed it in the manuscript accordingly:

As of now, there is a lack of prevalence data, estimates regarding at-risk populations, or any burden associated with Long COVID in Pakistan. This gap in knowledge primarily results from the absence of prospective longitudinal studies in the region.

Comment 7:

Page 3, Abstract: In the Results paragraph, it is written that “The most prominent symptoms that persisted even one year after infection…”. However, in the Conclusions paragraph, it is written that “This study reports the long-term health outcomes of COVID-19 survivors at 1-year following hospital discharge…”. The authors may wish to clarify when they assessed the study outcome; at one year after infection or one year after hospital discharge

Reply :

This has been rephrased in the manuscript, we did the study one year after discharge from the hospital.

Comment 8:

Page 6, Eligibility criteria and data collection, “Adult patients admitted between February 1, 2021, and June 7, 2021…”: In page 2, Abstract, it was written “…recruited patients aged > 18 years who were admitted between 1st January and June 30, 2021”. The authors may wish to correct such inconsistencies throughout their manuscript.

Reply :

We apologize for this error, this has been corrected in the manuscript.

Comment 9:

There are two Tables 6; one in page 18 and one in page 19.

Reply :

This has been corrected.

Comment 10:

Page 19, Figure 2 could be polished. For example, the authors could delete the “Chart title”.

Reply :

This has been changed.

Comment 11:

Pages 20-27: Discussion could be substantially shortened.

Reply :

As per your suggestion, we have shortened the discussion, from around 2416 words, we have reduced the discussion to 1995 words(pages 21-26)

---

## [Decision Letter · Decision Letter 1]

13 Oct 2023

PONE-D-23-22756R1Association between Long COVID and Vaccination: A 12-Month Follow-up Study in a Low- to Middle-Income countryPLOS ONE

Dear Dr. Ismail,

Thank you for submitting your manuscript to PLOS ONE. After careful consideration, we feel that it has merit but does not fully meet PLOS ONE’s publication criteria as it currently stands. Therefore, we invite you to submit a revised version of the manuscript that addresses the points raised during the review process.

There are clarifications and edits that need to be addressed.

We look forward to receiving your revised manuscript.

Kind regards,

Yatin N. Dholakia, MD

Academic Editor

PLOS ONE

Journal Requirements:

Additional Editor Comments:

There are some clarifications and edits that are requested which please address.

Reviewers' comments:

Reviewer's Responses to Questions

**Comments to the Author**

1. If the authors have adequately addressed your comments raised in a previous round of review and you feel that this manuscript is now acceptable for publication, you may indicate that here to bypass the “Comments to the Author” section, enter your conflict of interest statement in the “Confidential to Editor” section, and submit your "Accept" recommendation.

Reviewer #1: (No Response)

Reviewer #2: All comments have been addressed

2. Is the manuscript technically sound, and do the data support the conclusions?

Reviewer #1: Yes

Reviewer #2: Yes

3. Has the statistical analysis been performed appropriately and rigorously? 

Reviewer #1: Yes

Reviewer #2: Yes

4. Have the authors made all data underlying the findings in their manuscript fully available?

Reviewer #1: Yes

Reviewer #2: Yes

5. Is the manuscript presented in an intelligible fashion and written in standard English?

Reviewer #1: Yes

Reviewer #2: Yes

6. Review Comments to the Author

Reviewer #1: 1. Were long COVID symptoms the reason for re admission in any of the readmitted patients ?

2. “On multivariable analysis, after adjusting for age, gender, co-morbidity, and disease severity, vaccination was found to be an independent predictor of long COVID with an Odds ratio of 2.42(95% CI 1.52-3.84) for the unvaccinated cohort”.

This could probably be rephrased to lack of vaccination was found to be an independent predictor of long COVID……….

3. “Around 22 patients (4.58%) died after discharge from the hospital, however, it was difficult for us to assess the cause of death through telephonic interviews.”

a) Could you ascertain the point of time at which these deaths occurred….after 6 months or a year after discharge?

b) And, were the patients contacted anytime in between or directly after a year ?

c) Were the patients following up elsewhere during the year ?

d) Was imaging of the chest performed on OPD basis in any of the patients, to attribute the fatigue to post COVID sequelae like fibrosis ?

4. Taken together, it was observed that patients with severe/critical disease and unvaccinated status had an increased risk of long COVID and higher readmission rates than those in the non-severe and vaccinated cohort.

The above statement gives the impression that higher readmission rates were due to long COVID 19

5. Reported data regarding post-COVID sequelae one year after discharge from the hospital are similar and showed the persistence of symptoms in COVID-19 survivors…please clarify this statement

6. The discussion does appear long. Can be made crisper and to the point in order to hold the reader’s attention.

Reviewer #2: The authors addressed my comments.

7. PLOS authors have the option to publish the peer review history of their article (what does this mean?). If published, this will include your full peer review and any attached files.

Reviewer #1: **Yes: **Dr. Mala Kaneria

Reviewer #2: No

---

## [Author Response · Author response to Decision Letter 1]

7 Nov 2023

Thank you for your feedback on our revised manuscript (PONE-D-23-22756R1), titled “Association between Long COVID and Vaccination: A 12-Month Follow-up Study in a Low- to Middle-Income Country” at PLOS ONE. We are pleased to address the queries raised by Reviewer 1 and are equally delighted to learn that our response has met the satisfaction of Reviewer 2.

In response to Reviewer 1's thoughtful input, we have diligently refined our manuscript. These revisions have been made with great care to align our work with PLOS ONE's stringent publication standards. We eagerly await your assessment of the revised submission and remain at your disposal for any further questions or comments.

Yours sincerely,

On behalf of all the authors,

Dr. Madiha Ismail

Journal Requirements:

2. Additional Editor Comments:

There are some clarifications and edits that are requested, please address.

Reply:

● In response to your request for a review of our reference list to ensure its completeness and accuracy, we have thoroughly examined the references cited in our manuscript. We have checked and found that there were no retracted papers in our reference list. However, we have removed 2 references as we had to reduce the discussion as per the reviewer's advice.

• Sure we are providing the answer to the queries raised by Reviewer 1

"The following represents our response to the reviewers' comments."

Reviewer 1

Comment 1: 

Were long COVID symptoms the reason for re-admission in any of the readmitted patients?

Reply: 

We would like to express our gratitude to the reviewers for their insightful feedback. Regarding the question about whether Long COVID symptoms were the reason for re-admission in any of the readmitted patients, it's worth noting that while Long COVID symptoms were not documented as the principal or associated diagnosis in any of the discharge summaries/medical files, it is possible that some of the patients may be experiencing symptoms related to Long COVID at the time of readmission, but we did not find any documentation. We will incorporate this point in the manuscript.

Comment 2:

“On multivariable analysis, after adjusting for age, gender, co-morbidity, and disease severity, vaccination was found to be an independent predictor of long COVID with an Odds ratio of 2.42(95% CI 1.52-3.84) for the unvaccinated cohort”.

This could probably be rephrased to lack of vaccination was found to be an independent predictor of long COVID……….

Reply :

Thank you for the suggestion. The proposed rephrasing is more rational and we have incorporated this change in the revised manuscript using 'track changes’.

Comment 3: 

3. “Around 22 patients (4.58%) died after discharge from the hospital, however, it was difficult for us to assess the cause of death through telephonic interviews.”

a) Could you ascertain the point of time at which these deaths occurred….after 6 months or a year after discharge?

b) And, were the patients contacted anytime in between or directly after a year ?

c) Were the patients following up elsewhere during the year ?

d) Was imaging of the chest performed on an OPD basis in any of the patients, to attribute the fatigue to post COVID sequelae like fibrosis ?

Reply : 

The answers to comment 3 are given as follows:

a) Yes, we did examine the timing of these post-discharge deaths. The patients who expired following their discharge had a median survival duration of 31.5 days (IQR 7.5-213). 

b) The patients were contacted and interviewed one year after their discharge from the hospital. We did not contact them in between.

c) While some patients may have sought follow-up care at other healthcare facilities during the year following their discharge, however, we did not include this query as part of our interview.

d) Around 28.3% (n=136) of patients have chest X-rays done in the OPD (at any point from their hospital discharge up to the one-year interview). Fibrosis and consolidation were observed in 8.7% (n=42) and 7.7% (n=37) of the patients respectively, the rest of the chest X-rays were reported as normal. However, this percentage is underestimated as many of the patients may have chest X-rays done outside of our healthcare system. 

We have included these points in our manuscript and can be seen as track changes.

Comment 4: 

4. Taken together, it was observed that patients with severe/critical disease and unvaccinated status had an increased risk of long COVID and higher readmission rates than those in the non-severe and vaccinated cohort.

The above statement gives the impression that higher readmission rates were due to long COVID 19

Reply :

I appreciate your observation, and I can see how the statement may have given that impression. We have removed this statement and provided clarity wherever we have mentioned readmissions.

Comment 5: 

Reported data regarding post-COVID sequelae one year after discharge from the hospital are similar and showed the persistence of symptoms in COVID-19 survivors…please clarify this statement

Reply : 

We wanted to say that the data from across the world also show similar results but this statement was removed while we were revising our discussion.

Comment 6:

The discussion does appear long. Can be made crisper and to the point to hold the reader’s attention.

Reply :

We understand your advice for a more concise and focused discussion. So as per your recommendation, we have rewritten the discussion to hold the reader's attention.

---

## [Editor Report · Decision Letter 2]

9 Nov 2023

Association between Long COVID and Vaccination: A 12-Month Follow-up Study in a Low- to Middle-Income country

PONE-D-23-22756R2

Dear Dr. Madiha Ismail,

We’re pleased to inform you that your manuscript has been judged scientifically suitable for publication and will be formally accepted for publication once it meets all outstanding technical requirements.

Kind regards,

Yatin N. Dholakia, MD

Academic Editor

PLOS ONE
---

## [Editor Report · Acceptance letter]

13 Nov 2023

PONE-D-23-22756R2 

Association between Long COVID and Vaccination: A 12-Month Follow-up Study in a Low- to Middle-Income country  

Dear Dr. Ismail:

I'm pleased to inform you that your manuscript has been deemed suitable for publication in PLOS ONE. Congratulations! Your manuscript is now with our production department. 

Kind regards, 

on behalf of

Dr. Yatin N. Dholakia 

Academic Editor

PLOS ONE